# Enzymatic Synthesis of Poly(alkylene succinate)s: Influence of Reaction Conditions

Doris Pospiech [1,*], Renata Choińska [2], Daniel Flugrat [1,3], Karin Sahre [1], Dieter Jehnichen [1], Andreas Korwitz [1], Peter Friedel [1], Anett Werner [4] and Brigitte Voit [1,3]

[1] Leibniz-Institut für Polymerforschung Dresden e.V., Hohe Str. 6, 01069 Dresden, Germany; daniel.flugrat@mailbox.tu-dresden.de (D.F.); karin.sahre@primacom.net (K.S.); djeh@ipfdd.de (D.J.); korwitz@ipfdd.de (A.K.); friedel@ipfdd.de (P.F.); voit@ipfdd.de (B.V.)

[2] Prof. Waclaw Dabrowski Institute of Agricultural and Food Biotechnology—State Research Institute, Rakowiecka 36, 02-532 Warsaw, Poland; renata.choinska@ibprs.pl

[3] Organic Chemistry of Polymers, Technische Universität Dresden, 01069 Dresden, Germany

[4] Group Enzyme Technology, Bioprocess Engineering, Faculty of Mechanical Science and Engineering, Institute of Natural Materials Technology, Technische Universität Dresden, 01069 Dresden, Germany; anett.werner@tu-dresden.de

[*] Correspondence: pospiech@ipfdd.de; Tel.: +49-351-465-8497

**Abstract:** Application of lipases (preferentially *Candida antarctica* Lipase B, CALB) for melt polycondensation of aliphatic polyesters by transesterification of activated dicarboxylic acids with diols allows to displace toxic metal and metal oxide catalysts. Immobilization of the enzyme enhances the activity and the temperature range of use. The possibility to use enzyme-catalyzed polycondensation in melt is studied and compared to results of polycondensations in solution. The experiments show that CALB successfully catalyzes polycondensation of both, divinyladipate and dimethylsuccinate, respectively, with 1,4-butanediol. NMR spectroscopy, relative molar masses obtained by size exclusion chromatography, MALDI-TOF MS and wide-angle X-ray scattering are employed to compare the influence of synthesis conditions for poly(butylene adipate) (PBA) and poly(butylene succinate) (PBS). It is shown that the enzymatic activity of immobilized CALB deviates and influences the molar mass. CALB-catalyzed polycondensation of PBA in solution for 24 h at 70 °C achieves molar masses of up to $M_w\sim$60,000 g/mol, higher than reported previously and comparable to conventional PBA, while melt polycondensation resulted in a moderate decrease of molar mass to $M_w\sim$31,000. Enzymatically catalyzed melt polycondensation of PBS yields $M_w\sim$23,400 g/mol vs. $M_w\sim$40,000 g/mol with titanium(IV)n-butoxide. Melt polycondensation with enzyme catalysis allows to reduce the reaction time from days to 3–4 h.

**Keywords:** *Candida antarctica*; enzyme activity; enzymatic polycondensation; poly(butylene succinate); aliphatic polyester

## 1. Introduction

Enzymes are powerful biocatalysts enabling reactions in the human body, e.g., by generation of biopolymers. They catalyze highly efficiently and work under very mild conditions with respect to temperature, pressure, solvent and pH, and provide regioselectivity under suitable reaction conditions [1,2]. Therefore, efforts to use their catalytic activity for the preparation of synthetic polymers started several decades ago, in the eighties of the last century, and have been continuously increased year by year.

Numerous types of polymers prepared by enzymatic polymerization have been reported, among them polyphenols by enzymatic oxidative polymerization [3], polyaniline by enzymatic catalysis in presence of hydrogen peroxide [4], polysaccharides [5,6], vinyl polymerizates (being typically chain growth polymerizates) [7], a high number of different polyesters, polythioesters, polythioetheresters, polyphosphates, or polyketoetheresters obtained by ring-opening polymerization [8–11] or by polycondensation [12–14].

Polyesters and polyamides attracted particular attention, reflected by various studies [1, 15–24]. An advantage of enzymatic polycondensation is the possibility to polymerize monomers with reactive groups (e.g., oxirane and aromatic OH groups, double bonds like in itaconic acid [25–27], which would undergo side reactions under standard polycondensation conditions) resulting in polymers opening the opportunity for polymer-analogous reactions [12,15,25].

A comprehensive overview on enzyme types useful for catalysis of esterifications, among them oxidoreductases like laccase and horseradish peroxidase, transferases, hydrolases (in particular lipases), ligases, papain, trypsin, and α-chymotrypsin is provided in several reviews [15,17,25,28–30]. Lipase *Candida antarctica* Lipase B (CALB) occupies a prominent position because it also acts in organic solvents, covers a wide pH range, catalyzes a variety of organic reactions [31], but is very specific in esterification [17]. The biological function of lipase (triacylglycerol hydrolase) is the hydrolysis of triacylglycerol [22,32,33]. Conversely, it can also catalyze esterification. Figure 1 shows the schematic structure of *Candida antarctica* Lipase B and the active site for catalysis.

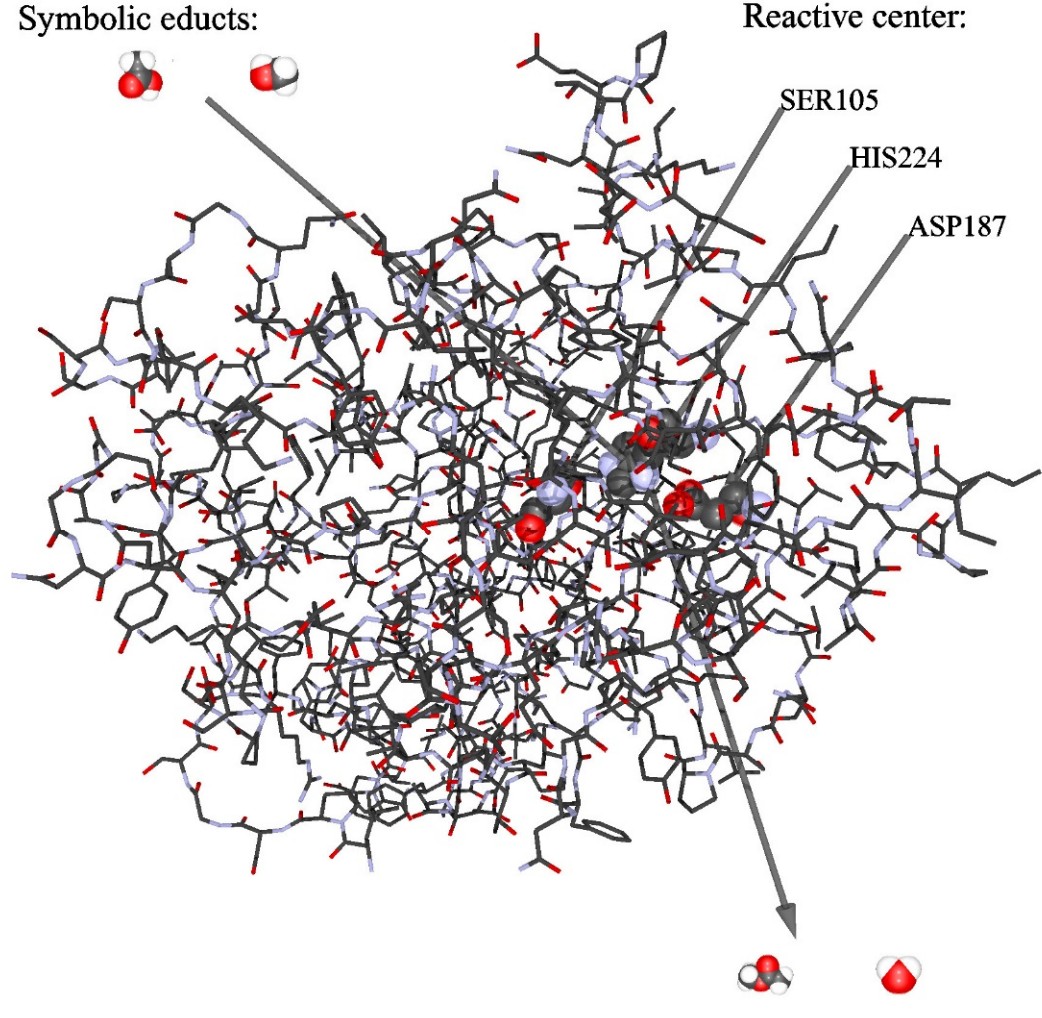

**Figure 1.** Schematic view on the active site of the enzyme *Candida antarctica* Lipase B. Illustration obtained with data given in [34] and program Rasmol (www.OpenRasMol.org, accessed on 15 January 2021).

The mechanism of the catalysis for transesterification is based on molecular recognition and has been described in detail variously [15,17,25,33]. Considering this mechanism, it becomes obvious that only the reaction of substrates fitting into the active site of the

enzyme can be catalyzed (see also Figure 1). Most suitable substrates for enzymatic poly-condensation belong to the groups of aliphatic diols and aliphatic dicarboxylic acids owing to the hydrophobic nature of the methylene groups positioned between the functional groups. A large variety of aliphatic polyesters has already been synthesized by enzymatic polycondensation under different conditions [1,12,17,33]. It was noticed that maximum molar masses were obtained with 1,6-hexanediol and adipic/sebacic acid [17], reflecting the importance of the hydrophobicity of the reactants.

The manifold attempts to use lipase-catalyzed polycondensation mainly with CALB N435 for aliphatic polyesters are briefly summarized in the following. In most of the studies, only low molar masses were obtained even after long reaction times. The relative molar masses given should not be compared directly because they were obtained under different lab conditions. The values only give a rough trend. Chaudhary et al. [18–20] used solvent-free polycondensation of divinyl adipate and 1,4-butanediol and achieved maximum averaged relative molar masses $M_w$ of about 23,000 g/mol with N435 at 50 °C. Mahapatro et al. [35] reported $M_w$'s up to 25,000 g/mol for bulk direct polycondensations with 1,8-octanediol at 70 °C and long reaction times (50 h). Jiang et al. [15] reported either melt polycondensation of succinic anhydride and 1,4-butanediol catalyzed by N435 at 95 °C for 12 h, or polycondensation of dimethyl succinate and 1,4-butanediol in bulk at 120 °C for 24 h to obtain polyesters with high molar mass. Ren et al. [36] used a two-step process involving bulk polymerization of the acid anhydride and the diol in the first step, followed by solution polycondensation in toluene with enzyme for 30 h. Another two-step polycondensation with a 24 h step under vacuum was applied by Nasr et al. [37] yielding molar masses $M_n$ of ~12,000 g/mol in maximum. Molar masses of further two-step polycondensation for poly(butylene succinate) (PBS) were not given [38]. High molar mass PBS was obtained after removal of condensation products by azeotropic polymerization. Azim et al. [39] reported CALB-catalyzed copolymerization of diethyl succinate with 1,4-butanediol in bulk and in solution PBS with $M_n$ = 3300 g/mol in bulk versus PBS with $M_n$ = 10,000 g/mol in diphenyl ether was obtained. The lipase-catalyzed polycondensation reactions under solvent-less conditions have been carried out also for synthesis of polyesters from adipic acid and sebacic acid with different diols [12,40–43] yielding different molar masses from $10^3$–to $10^4$ g/mol. Aliphatic building blocks with unsaturated units can also be converted without crosslinking [25,44–48]. Aliphatic-aromatic polyesters were prepared enzymatically as well [49]. Here, often highly activated divinyl derivatives of the dicarboxylic acids were used, still resulting in low molar mass polymers. In most cases, CALB immobilized to crosslinked polymeric supports or natural products was utilized because immobilization usually improves the resistance of the enzyme in organic solvents, the long-term activity and thermal stability [25,28,44,50–53]. One of the most prominent catalysts of this type is Novozyme 435 (N435) [15], which was also used in this study.

Despite the huge progress made for CALB-catalyzed polycondensation reactions, further studies intensifying the knowledge in this field are necessary. In particular, more re-search has to be done for commercial implementation of lipase-catalyzed polymerization in bulk. There is a growing interest in the development of environmentally compatible processes, and lipase-catalyzed polycondensation in bulk becomes more eco-friendly over that performed in solvents.

In this study, we investigated the applicability of enzymatic catalysis by CALB to melt polycondensation for preparation of poly(butylene adipate (PBA) and PBS. Biodegradabil-ity, relatively good mechanical properties and processability make these aliphatic polyesters useful for a variety of industrial, medical, and agricultural applications, e.g., in different areas ranging from food packaging, fiber/textile, electrics and electronics, and agricultural mulch films [54–57]. Melt polycondensation is the most important method for polyester synthesis [58]. The commercial synthetic route to PBS employs melt polycondensation of 1,4-butanediol and succinic acid, catalyzed by organometal or metal oxide catalysts [59]. High molar masses can be reached, but disadvantages are the toxic catalysts, the occurrence

of side reactions and rather harsh reactions conditions with reaction temperatures close to thermal degradation. Therefore, usage of enzymatic catalysis is highly appreciated. Here, series of PBS and PBA polyesters were synthesized by CALB–catalyzed polycondensation in bulk as well as in solution, and by standard transesterification polycondensation in bulk. Dimethyl succinate and 1,4-butanediol were used for the synthesis of PBS. Divinyl adipate and 1,4-butanediol were chosen for the synthesis of PBA. The main purpose was to find out whether it is possible to replace conventional titanium(IV)n-butoxide catalysts by CALB without drastic change of reaction conditions and to prepare polyesters with sufficiently high molar mass, and if possible with melt polycondensation. Both, CALB-catalyzed and standard polycondensations were carried out under defined and, if possible, comparable reaction conditions. This attempt allowed studying the effect of CALB activity and used reaction conditions on the relative molar mass of the synthesized polyesters. The formed products were characterized in detail by SEC, $^1$H NMR, MALDI-TOF MS, and WAXS analysis.

## 2. Materials and Methods

### 2.1. Materials

The immobilized enzyme *Candida antarctica* Lipase B (CALB, Novozyme 435, N435) was obtained by Novozymes (Bagsværd, DK). It can also be supplied by Sigma-Aldrich (Sigma-Aldrich Chemie GmbH, Munich, Germany). N435 is CALB adsorbed by interfacial activation to a macroporous acrylic polymer network [52]. Divinyl adipate (>99.0%, TCI Deutschland GmbH, Eschborn, Germany), dimethyl succinate (DMS, 98%, Sigma-Aldrich Chemie GmbH, Munich, Germany), dimethyl adipate (DMA, 99%, Sigma-Aldrich Chemie GmbH, Munich, Germany), toluene (99.85%, ACROS Organics, Thermo Fisher, Schwerte, Germany) and 1,4-butanediol (99%, Sigma-Aldrich Chemie GmbH, Munich, Germany) were stored over molecular sieves. Titanium(IV)n-butoxide (Ti(OBu)$_4$, 99%, ACROS Organics, Thermo Fisher, Schwerte, Germany) and antimony trioxide (Sb$_2$O$_3$, 99.9%, Sigma-Aldrich Chemie GmbH, Munich, Germany) were dried in a vacuum oven for 4 h at 100 °C prior to polycondensation. Chloroform (99%, ACROS Organics, Thermo Fisher, Schwerte, Germany), methanol (99%, ACROS Organics, Thermo Fisher, Schwerte, Germany), pentafluorophenol and the chemicals for the enzyme assay, *p*-nitrophenyl palmitate (lipase substrate, Sigma-Aldrich Chemie GmbH, Munich, Germany), sodium desoxycholate (BioXtra, ≥98.0%, Sigma-Aldrich Chemie GmbH, Munich, Germany), Triton X-100 detergent (Thermo Fisher, Schwerte, Germany), Gum arabic (Sigma-Aldrich Chemie GmbH, Munich, Germany) and 2-amino-2-hydroxymethyl-propane-1,3-diol (tris) buffer (100 mmol pH 7.4, Molecular Biology Grade, Merck KgaA, Darmstadt, Germany) were used as received.

### 2.2. Procedures

Control polycondensations with Ti(OBu)$_4$ (sample PBA9, 1 wt.-%), and for PBS1 with Ti(OBu)$_4$ and Sb$_2$O$_3$ as catalyst mixture (0.5./0.5 wt.-% with respect to the total weight of monomers) were carried out as melt polycondensation using dimethyladipate or dimethylsuccinate, respectively, as reported in [60]. Thus, the dimethylester (0.04 mol) and 1,4-BD (0.08 mol) were added in a three-necked flask equipped with nitrogen inlet, mechanical stirrer and distillation head. After three times evacuation followed by purging with nitrogen the flask was inserted into a heating bath at 150 °C where the reaction started. The polycondensation proceeded under stirring, nitrogen flow at 245 °C and heating to 245 °C within 60 min. At 245 °C, the mixture was stirred for further 30 min at 245 °C. Finally, the polycondensation was carried out under vacuum ($1.7 \times 10^{-1}$ mbar) at 245 °C for 3 h. Then, the flask was removed from the heating bath, cooled down and the product was removed from the flask.

The enzyme-catalyzed polycondensation of divinyl adipate with 1,4-butanediol was performed according to the procedure described by Dai et al. [61]. The activity of the CALB used is given in Table 1. The molar ratios of monomers and the solvent concentrations used are given in Table 2. CALB (10 wt.-% with respect to the total weight of monomers, for all

reactions), 1,4-butanediol and toluene (without solvent in case of bulk polymerization) were given into a 50 mL round–bottom flask equipped with magnetic stirrer, nitrogen inlet and condenser. The reaction flask was placed into an oil bath with constant preset temperature and the mixture was stirred at 70 °C under nitrogen for 1 h. Then, an aliquot of vinyl adipate was added and the reaction mixture was further stirred, under the same conditions, for a predetermined time, see Table 1. The reaction was stopped by addition of an excess of chloroform. Enzyme beads were filtered off, chloroform was removed under vacuum and the product was obtained upon precipitation into the tenfold excess of cold methanol. After filtration, the white powder was dried in a vacuum oven at 40 °C for 24 h. Yields: see Table 2. $^1$H NMR: (CDCl$_3$): δ (ppm): PBA: 1.69 (8H, C(=O)CH$_2$CH$_2$ and OCH$_2$CH$_2$); 2.32 (4H, C(=O)CH$_2$); 4.08 (4H, OCH$_2$). PBS: 1.71 (4H, -CH$_2$-); 2.62 (4H, -COCH$_2$-), 4.12 (4H, -CH$_2$O-).

**Table 1.** Activities of the *Candida antarctica* Lipase B (CALB) samples N435 employed as catalyst for polycondensation.

| Sample | Description | $A$ at 50 °C (U/g) |
|---|---|---|
| CALB1 | Stored 2 years | $1400 \pm 100$ |
| CALB2 | Stored 6 months | $3600 \pm 300$ |
| CALB3 | Fresh | $2900 \pm 700$ |
| CALB1r | Recovered after solution polycondensation at 80 °C | 500 |

**Table 2.** Polycondensation results for the synthesis of PBA using divinyladipate and 1,4-butanediol, and PBS using dimethylsuccinate and 1,4-butanediol.

| Entry | Ratio Ester/BD (mmol/mmol) | Cat. | Ratio mon./toluene (g/mL) | $t$ (h) | $T$ (°C) | $M_{n,SEC}$ (g/mol) | $M_{w,SEC}$ (g/mol) | $Đ = M_w/M_n$ | $\eta_{inh}$ (dL/g) | Yield (%) |
|---|---|---|---|---|---|---|---|---|---|---|
| PBA1 | 4.5/4.5 | CALB1 (dried) | 0.259 | 8 | 70 | 1400 | 4700 | 3.36 | n.d. | 43 |
| PBA2 | 4.5/4.5 | CALB1 | 0.259 | 8 | 70 | 1600 | 5000 | 3.12 | n.d. | 52 |
| PBA3 | 4.5/5.0 | CALB1 | 0.268 | 8 | 70 | 6400 | 9700 | 1.52 | 0.27 | 39 |
| PBA4 | 4.5/5.0 | CALB1 | 0.268 | 24 | 70 | 18,300 | 59,500 | 3.25 | 0.79 | 37 |
| PBA5 | 4.5/5.0 | CALB2 | 0.447 | 24 | 70 | 12,300 | 35,300 | 2.87 | 0.65 | 69 |
| PBA5-1 | 4.5/5.0 | CALB2 | 0.447 | 8 | 70 | 8100 | 14,800 | 1.84 | n.d. | 42 |
| PBA6 | 4.5/5.0 | CALB2 | melt | 24 | 70 | 7100 | 12,600 | 1.78 | n.d. | 33 |
| PBA7 | 9.0/9.0 | CALB2 | melt | 24 | 70 | 7100 | 31,000 | 4.37 | 0.41 | 74 |
| PBA8 | 4.5/5.0 | P.F. [b] | 0.447 | 24 | 70 | no react. | - | - | - | - |
| PBA9 [a] | 4.5/4.5 | Ti(OBu)$_4$ | melt | 4 | 150–245 | 28,000 | 58,000 | 2.12 | 0.70 | 90 |
| PBS1 | 4.5/4.5 | Ti(OBu)$_4$ | Melt(vac) | 3 | 245 | 40,500 | 93,000 | 2.29 | 0.90 | 92 |
| PBS1-2 | 4.5/4.5 | Ti(OBu)$_4$ | Melt(vac) | 4 | 245 | 18,500 | 40,000 | 2.16 | 0.55 | 90 |
| PBS1-3 [c] | 4.5/4.5 | Ti(OBu)$_4$ | Melt(vac) | 4 | 245 | 147,000 | 308,000 | 2.09 | 1.8 | 89 |
| PBS2 | 4.5/4.5 | CALB1 | 0.044 | 24 | 70 | 2900 | 3800 | 1.31 | n.d. | 51 |
| PBS3 | 4.5/4.5 | CALB2 | 0.044 | 24 | 70 | 5400 | 8400 | 1.56 | n.d. | 53 |
| PBS4 | 4.5/4.5 | CALB3 | 0.044 | 24 | 70 | 3800 | 5200 | 1.37 | n.d. | 50 |
| PBS5 | 4.5/4.5 | CALB3 | 0.044 | 24 | 80 | 11,000 | 21,300 | 1.93 | 0.31 | 56 |
| PBS6 | 4.5/4.5 | CALB3 | 0.213 | 24 | 70 | 5400 | 7100 | 1.31 | 0.87 | 40 |
| PBS7 | 4.5/4.5 | CALB2 | melt | 1 | 80 | 4300 | 5300 | 1.23 | 0.17 | 40 |
| PBS8 | 4.5/4.5 | CALB2 | melt | 1.50 0.25 | 80 150 | 11,000 | 19,700 | 1.79 | 0.35 | 75 |
| PBS9 | 4.5/4.5 | CALB2 | melt | 1.50 0.25 | 80 200 | 3900 | 5400 | 1.38 | 0.16 | 42 |
| PBS10 | 4.5/4.5 | CALB3 | Melt (vac) | 0.75 3.00 | 80 200 | 4600 | 6450 | 1.40 | 0.17 | 45 |
| PBS11 | 4.5/4.5 | CALB3 | Melt (vac) | 0.75 3 | 80 130 | 11,700 | 23,600 | 2.02 | 0.36 | 75 |

[a] Polycondensation with dimethyl adipate instead of divinyl adipate; [b] PF: *Pseudomonas fluorescencis;* [c] upscaling in 2.4 L-stirring autoclave; n.d.: Not determined; no react.: Polycondensation did not occur.

The enzymatic polycondensation of dimethyl succinate with 1,4-butanediol in melt was carried out in a 50 mL three-necked flask equipped with mechanical stirrer, vacuum line

and nitrogen inlet. The monomers DMS (9 mmol) and 1,4-butanediol (9 mmol) as well as the immobilized enzyme (10 wt.-% with respect to the monomer weight) were given into the flask. The flask was secured three times by applying vacuum followed by nitrogen purge. The flask was inserted into an oil bath pre-heated to 80 °C and the polycondensation was carried out under stirring for 45 min at that temperature. Then, the temperature was gradually raised to 120, 130, and 150 °C, respectively, where the reaction was performed for further 3 h under reduced pressure ($1.7 \times 10^{-1}$ mbar). The flask was removed from the bath and cooled down. Then, the product was dissolved with chloroform (10 mL) and the residual enzyme was filtered off. The polymer was precipitated from chloroform into the ten-fold amount of methanol, filtered and dried under reduced pressure for 24 h at 50 °C. Yield: 90%.

### 2.3. Methods

#### 2.3.1. Determination of Enzyme Activity

The activity of CALB was determined using the simplified *p*-nitrophenyl palmitate assay for lipases and esterases as reported by Gupta et al. [62]. The reaction mixture was freshly prepared before each measurement and consisted of solution 1 (1 mL; 30.3 mg *p*-nitrophenyl palmitate in 10 mL isopropanol) and solution 2 (9 mL; consisting of tris buffer (3.057 mg) in milliQ water (0.5 mL) at a pH of 8.0, Triton X-100 (2 mL), sodium desoxycholate (1003 mg) and Gum arabic (503 mg)). The immobilized enzyme (1 mg) was given into a 1.5 mL Eppendorf vial and was subjected to a constant temperature (usually 50 °C) in a Thermomix (Eppendorf, Germany). After reaching constant temperature the reaction mixture (0.6 mL, pre-heated to 50 °C) was added and the reaction time started within the Thermomix with a rotational speed of 1000 rpm. A control sample without enzyme was included as blank value. Samples were taken after 5 and 10 min and the reaction was stopped by adding cold ethanol (0.5 mL) and immediate cooling of the vial. After cooling down, the solution was centrifuged at 3500 rpm at 4 °C for 5 min. Then, the solution was immediately examined in an UV-Vis spectrometer (Beckman DU640, Germany) at 405 nm. The activity *A* was calculated according to the following equation:

$$A = (\Delta E \cdot V)/(\Delta t \cdot \varepsilon \cdot d) \tag{1}$$

with $\Delta E$ being the difference between the extinction of sample and control, *V* being the sample volume, $\Delta t$ the difference between the measuring times, *d* as thickness of the cuvette, and $\varepsilon$ being the extinction coefficient of p-nitrophenol ($2.71 \times 10^3$ mol$^{-1}$cm$^{-1}$). The values were averaged from at least two measurements with the deviations of 20–37%.

#### 2.3.2. Solution Viscosity

The solution viscosity of the polymers was determined in an Ubbelohde viscometer AVS 470 (SI Analytics GmbH, Mainz, Germany) using capillary 537 10/I. The polymer was dissolved in a mixture of pentafluorophenol/chloroform (50/50 *vol/vol*) with a concentration of 5 g/L. After filtration through a PTFE syringe filter (pore size 0.2 μm) the solution was inserted into the capillary. The measurements were performed at 25 °C. The solution viscosity $\eta_{inh}$ was determined according to Equation (2):

$$\eta_{inh} = \ln(\eta_{rel})/c \tag{2}$$

with polymer concentration *c* in dL/g and $\eta_{rel}$ being the ratio of the flow time of the polymer solution to the flow time of the solvent. The values were determined as average of five measurements.

#### 2.3.3. Size Exclusion Chromatography

The relative molar masses of the polymers were determined by size exclusion chromatography (SEC) employing an HPLC pump Series 1200 (Agilent Technologies, Santa Clara, CA, USA) equipped with differential refractometer (Knauer, Berlin, Germany) and sepa-

ration columns PL MiniMix-D (250 × 4.6 mm) and PSgel (5 µm) (both Agilent Technologies, Santa Clara, CA, USA). The eluent was a mixture of pentafluorophenol/chloroform (33/67 *vol/vol*) and the flow rate was 0.3 mL/min. The measurements were performed at 45 °C. The relative molar masses were calculated with respect to PS standards (Easi Vial PS, PSS GmbH, Mainz, Germany).

### 2.3.4. NMR Spectroscopy

$^1$H NMR was carried out in deuterated chloroform (CDCl$_3$). The spectra were recorded with a Bruker Avance III 500 spectrometer (Bruker Biospin, Billerica, MA, USA) at 500.13 MHz. The spectra were referenced to the signal at 7.26 ppm.

### 2.3.5. Matrix-Assisted Laser Desorption Ionization–Time-of-Flight Mass Spectrometry (MALDI-TOF MS)

The MALDI-TOF experiments were performed on an Autoflex Speed TOF/TOF system (Bruker Daltonics Rosenheim, Germany). The measurements were carried out in a linear mode and positive polarity by pulsed smart beam laser (modified Nd:YAG laser). The ion acceleration voltage was set to 20 kV. For the preparation of PBA, the samples were mixed with 2,5-dihydroxy benzoic acid as matrix, both were dissolved in tetrahydrofuran. PBS samples were prepared by mixing the polymers with dithranol as matrix in chloroform. Sodium trifluoroacetate was added in all preparations.

### 2.3.6. Differential Scanning Calorimetry (DSC)

The DSC measurements were performed on a Q 2000 (TA Instruments, Newcastle, DE, USA) in a temperature range from −80 to 200 °C at a scan rate of ±10 K/min with nitrogen as pure gas. The first heating, cooling, and second heating run were recorded. 5 mg of sample was used.

### 2.3.7. Wide-Angle X-ray Scattering (WAXS)

The measurements were carried out with the as-synthesized, dried and powdered samples. The solid state structure (crystalline modifications α and β) was determined by wide-angle X-ray scattering (WAXS) with a XRD 3003 T/T (GE Sensing & Inspection Technologies GmbH, Ahrensburg, Germany) with 40 kV/30 mA CuKα radiation ($\lambda$ = 0.1542 nm, monochromatic with primary multilayer system) in the measuring range of $2\theta$ = 0.5–40° in symmetric transmission and step-scan mode ($\Delta 2\theta$ = 0.05°) with a measuring time of 15 s for each point. The primary data (intensity $I$) were presented as radial profiles $I(2\theta)$. The Bragg's law [63] is the basic equation for scattering experiments (Equation (3)):

$$2d \cdot \sin \theta = n \cdot \lambda \tag{3}$$

with $d$—lattice plane distance, $\theta$—half scattering angle, $n$—order of reflection, and $\lambda$—wavelength. The total crystallinity $c_{X,\text{total}}$ of the samples was determined according to the peak area method [64] with Equation (4):

$$c_{X,\text{total}} = c_{X,\alpha} + c_{X\beta} \tag{4}$$

with $c_{X,\alpha}$ as crystallinity of the α phase and $c_{X\beta}$ as crystallinity of the β phase. The calculations were performed using underground-corrected WAXS curves in which all available scattering maxima (reflections and amorphous halo inside the appropriate scattering range ($2\theta$ = 7–37°) are fitted using pseudo-Voigt functions with the software tool "Analyze" of the device software package "RayfleX" for X-ray scattering devices (GE Sensing & Inspection Technologies GmbH, Ahrensburg, Germany). A small number of sharp reflections caused by low amounts of the catalyst antimony trioxide ($2\theta$: 13.7, 27.7, 32.1, and 35.2°; e.g., in sample PBS1) were corrected before calculation. The errors of the crystallinities were $\Delta c_{X,\text{ total}} \approx \pm 0.01$ and $\Delta c_{X,\text{ α or β}} \approx \pm 0.02$, respectively.

## 3. Results and Discussion

### 3.1. Activity of the Lipases

The polymer-supported CALB N435 was chosen for the study because it has frequently been mentioned as successful catalyst for enzymatic polycondensation [15,39,45,65]. Three samples of immobilized CALB N435 with different age were employed in the study. Their activity was measured at 50 °C taking values in the linear time range at 5 and 10 min. The results in Table 1 show that the activities of the samples at 50 °C strongly depended on the storage time after purchase. One sample was recovered after polycondensation in toluene. A remaining activity even if reduced compared to the fresh sample was proven. Measurements at 80 °C to estimate the activity at the temperature of solution polycondensations failed and did not give meaningful values.

### 3.2. Polycondensation Results

The synthesis of PBA and PBS catalyzed by the immobilized CALB with activities given in Table 1 was performed with activated esters and free diols according to the reaction equation in Figure 2. Divinyladipate was used as activated form of adipic acid and dimethyl succinate as activated form of succinic acid to avoid side reactions occurring in polycondensations with free dicarboxylic acids. The main aim of the study was directed to the question whether or not the reaction can be carried out without organic solvent as melt polycondensation and, in case of PBS, at higher reaction temperatures than reported in the literature. It is known that the activity of enzymes is strongly reduced at higher temperature. Thus, a deactivation under the conditions employed had to be expected.

R = vinyl, m = 2: divinyladipate    - PBA
R = CH$_3$, m = 1: dimethylsuccinate    - PBS

**Figure 2.** Reaction scheme for the enzymatic polycondensation of dicarboxylic esters with 1,4-butanediol.

The results of the experiments with respect to molar mass, molar mass distribution and solution viscosities (selected samples) are summarized in Table 2. In the discussion of the results and especially in the comparison with literature results it has to be taken into account that the molar masses are relative values and may also depend on the analysis conditions. Therefore, all samples analyzed here were examined under comparable conditions. Furthermore, it has to be taken into account that the melt polycondensates obtained enzymatically were dissolved and re-precipitated to remove N435 for SEC analysis, resulting in a reduction of yield of about 10 wt.-% and change of molar mass after polycondensation.

The polycondensation of divinyladipate with 1,4-butanediol was studied first taking literature reports into account stating that enzymatic polycondensation with diethyladipate yielded only low molar mass products (below 2000 g/mol) [66,67]. The vinyl group is a more reactive leaving group for transesterification than the methoxy group in dimethylsuccinate. The vinyl alcohol which is generated during reaction tautomerizes to acetaldehyde and evolves into the gas phase due to the low boiling point of acetaldehyde (20.2 °C), rendering the reaction irreversible and shifting the equilibrium towards the reaction product [12]. The conditions were chosen similar to those reported by Dai et al. [61] of the polycondensation of divinyladipate with polyester macrodiols. The solution polycondensation was carried out in toluene at 70 °C under nitrogen flow. Headspace gas chromatography of the distillates collected during polycondensation indeed revealed formation of acetaldehyde (see Supporting Information, Figure S1). Drying N435 did not influence the result in solution (sample PBA1 vs. PBA2). The relative molar masses resulting after 8 h reaction time

were low ($M_w$ ~5000 g/mol) and in the range of literature reports [68,69]. Slight excess of 1,4-butanediol enhanced the molar mass (samples PBA2 vs. PBA3, $M_w$ ~10,000 g/mol). Extending the reaction time from 8 h to 24 h resulted in a significant increase of relative molar mass to $M_w$ ~60,000 g/mol (sample PBA3 vs. PBA4). Reducing the monomer concentration in toluene (PBA4) under the same conditions, but with CALB2 with higher activity, reduced the molar mass to $M_w$ ~35,000 g/mol. This is in agreement with the theory of polycondensation [70]. The comparison of CALB1 and CALB2 under similar conditions resulted in higher relative molar mass for the sample PBA5-1 obtained with CALB2 with higher enzymatic activity (increase of molar mass from $M_w$ ~ 10,000 g/mol for PBA3 to ~15,000 g/mol for PBA5-1). Polycondensation in bulk (melt) at the same temperature (80 °C) reduced the molar mass (samples PBA6 and PBA7) due to the diffusion control of the reaction. The change of the enzyme to *Pseudomonas fluorescencis*, which was reported in the literature as catalyst [51], was not successful and a polymer was not formed (PBA8). Melt polycondensation of the less reactive dimethyladipate and 1,4-butanediol under standard transesterification conditions with titanium(IV)n-butoxide at 245 °C for 4 h (sample PBA9) yielded an $M_w$ ~60,000 g/mol, which is comparable to the sample obtained with divinyladipate in solution (PBA4). Enzymatically catalyzed melt polycondensates (PBA6, PBA7) had lower molar masses, despite the fact that the melting temperature of PBA is below the reaction temperature in the range of 60 °C [71,72].

　　MALDI-TOF MS of samples PBA3 and PBA6 proved the formation of higher molecular species with the expected repeating unit of PBA with a molar mass of 200.2 Da. Figure 3 illustrates the MALDI-TOF spectrum of PBA6 obtained with the matrix dihydroxybenzoic acid in linear mode. The spectrum reveals a regular molar mass distribution. The distributions of the existing species have a repeating unit of 200.7 Da consistent with PBA. The major distribution (peaks with highest intensity) originates from polyesters with two vinylester end groups, the distribution with EG 17/1 has a carboxylic and an H end group, the distribution with 43/1 had both a vinylester and an H end group, and the distribution with the lowest intensity is caused by a minor content of cyclic structures (without end groups). Thus, MALDI-TOF MS proved that enzymatic polycondensation of divinyladipate and 1,4-butanediol worked properly.

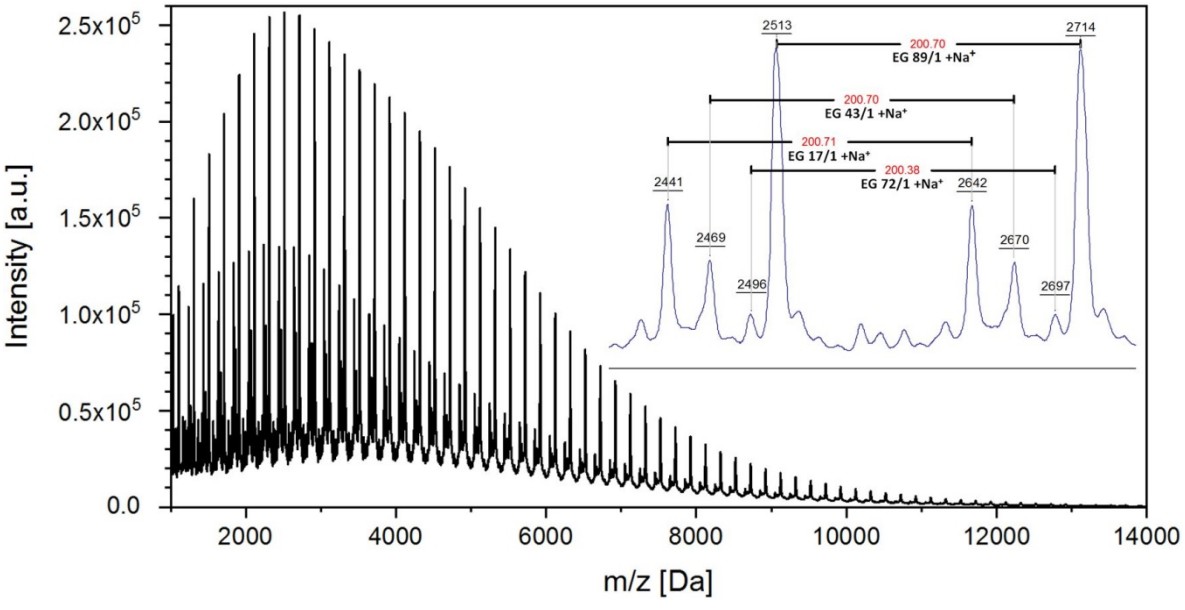

**Figure 3.** MALDI/TOF-MS spectrum of PBA6 (EG: end group).

　　Both, molar mass and MALDI-TOF MS results were supported by the ¹H NMR spectra of the samples (not shown here because the signal assignments are well-known from literature [61,71,72]). High molar mass samples, e.g., PBA4 with $M_w$~35,000 g/mol

showed only the signals assigned to the polymer repeating units (Figure S2), while samples with lower molar mass showed vinylester end groups at 4.88/4.86 and 4.57/4.56 ppm (sample PBA2, Figure S5), and those with intermediate molar mass OH end groups (PBA5, PBA5-1, PBA7) signals at 3.68 ppm (Figures S3 and S4) which can be explained by the slight excess of 1,4-butanediol used.

In the next step, the reaction between dimethyl succinate and BD under varying reaction conditions was examined. Here, the dimethyl ester of succinic acid was employed to adjust the conditions as close as possible to the conditions of the standard polycondensation with titanium(IV)n-butoxide as catalyst. Solution polycondensation in toluene at 70 °C under comparable conditions yielded PBS with $M_w$ below 10,000 g/mol (PBS1, PBS2, PBS3). The influence of enzyme activity can clearly be detected: The higher the activity, the higher is the relative molar mass (see Table 2). Increasing the temperature to 80 °C enhanced the resulting $M_w$ to 21,300 g/mol (PBS5).

However, the main focus was on using melt polycondensation. The materials obtained by melt polycondensation were dissolved to remove the insoluble enzyme-immobilized resin by filtration, and then the polymer was precipitated and dried for analysis of molar masses. This procedure normally results in reduced relative molar masses compared to those without work-up (which is the normal state in which the materials will be used in practice). Thus, higher molar masses of the materials than actually determined by SEC can be assumed in the materials which are not worked-up.

Standard polycondensation was performed at 245 °C (samples PBS1 and PBS1-2 obtained in lab-scale and PBS1-3 in a 2.4 L-stirring autoclave. The relative molar mass of the up-scaled sample PBS1-3 is in the range of that determined for commercial PBS Bionolle® ($M_w$ ~ 308,000 g/mol vs. $M_w$ ~ 355,000 g/mol for Bionolle®). These were the highest molar masses of the PBS series. It was shown before that the temperature of 245 °C is appropriate because the main thermal decomposition of PBS takes place in the range of 400 °C [73]. Conducting the polycondensation without any catalyst at 200 °C did not result in polymer formation. A low molar mass polymer was formed when the reaction was carried out in melt at 80 °C for one hour (PBS7). This is due to the fact that the melting range of PBS is above 100 °C (see Table 2). The reaction became diffusion controlled and stopped when polymer chains with melting temperatures higher than the reaction temperature were formed. Therefore, the reaction temperature was stepwise enhanced in the experiments from 80 °C to 200 °C (in the first phase, the temperature remained constant at 80 °C and was then quickly enhanced to the second polycondensation temperature, see Table 2). It was assumed that the enzyme activity quickly decreases at higher temperature (activity measurements at $T > 50$ °C failed). Using 200 °C for the second polycondensation phase (samples PBS9 and PBS10) yielded low molar mass polymers indicating that the enzyme is not active anymore at this temperature, or, alternatively, catalyzes the polymer degradation. A reaction temperature between 130 and 150 °C (samples PBS11 and PBS8) was optimal and resulted in $M_w$ up to 23,600 g/mol comparable to the best solution polycondensate PBS5, but lower than the titanium-catalyzed melt polycondensates PBS1 and PBS1-2.

The polymers were analyzed additionally by MALDI/TOF-MS in combination with [1]H NMR. The MALDI/TOF-MS spectra are displayed in Figure 4. First of all, only low molar parts of the samples up to masses of about 4000 Da, far below the masses found by SEC, can be detected by MALDI/TOF-MS. That means, high molar mass linear PBS is not vaporized and detected under the conditions used (even though different conditions were checked). A comparable result was reported by Garaleh et al. [74]. The complete sample is only represented by the [1]H NMR spectrum. In all MALDI/TOF-MS spectra, the repeating unit characteristic for PBS (172.8 Da) was identified. Sample PBS1-2 with intermediate molar mass obtained with conventional titanium(IV)n-butoxide catalyst showed one main mass distribution of polymers without end groups. These species can be assigned to cyclic oligomers up to 4000 Da. The high molar mass part of the sample could not be vaporized and not detected. The cyclic polymers can be formed by thermal conversion of butylene hydroxide end groups into vinylester end groups (indeed detected in the [1]H NMR

spectrum, Figure S6) and split-off these groups. The enzyme-catalyzed low molar mass sample PBS7 is characterized by several mass distributions. The main distribution belongs to polymers terminated with both kinds of end groups (methylester and OH, visible in the [1]H NMR spectrum Figure S7 by signals at 3.36 ppm and at 3.67 ppm), but also cyclic oligomers are found. The main distribution of the higher molar mass sample PBS5 can be assigned to polymers with OH/H end groups (i.e., terminated at both ends with OH groups, Figure S8).

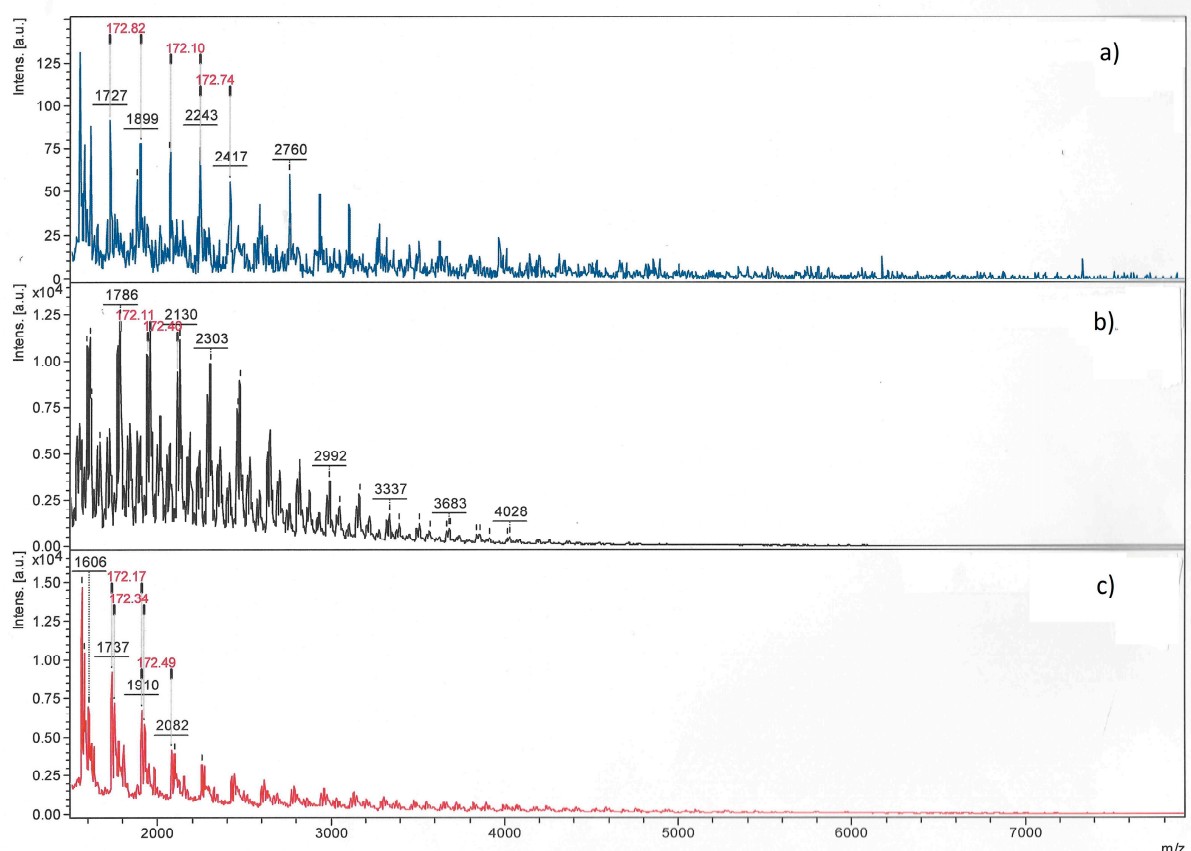

**Figure 4.** MALDI/TOF-MS spectra of PBS (**a**) PBS1-2 obtained by conventional transesterification with Ti(OBut)$_4$; (**b**) PBS7 obtained with CALB in melt with low molar mass, and (**c**) PBS5 obtained with CALB in solution with high molar mass.

In summary, conditions for melt polycondensation of PBS and PBA resulting in materials with sufficiently high molar masses could be identified in the experiments.

### 3.3. Influence of Polycondensation Conditions on the Solid State Structure

Aliphatic polyesters are semicrystalline materials with a usually distinct crystalline phase [13,35,36,75].

Ichikawa et al. [76–78] reported for PBS the occurrence of both $\alpha$- and $\beta$-modification with monoclinic cells. The $\beta$-form was observed after application of stress. For the monoclinic $\alpha$-modification, the cell dimensions were reported to be $a = 0.523$ nm, $b = 0.912$ nm, $c$ (fiber axis) $= 1.090$ nm, and $\beta = 123.9°$; for the $\beta$-form to be $a = 0.584$ nm, $b = 0.832$ nm, $c$ (fiber axis) $= 1.186$ nm, and $\beta = 131.6°$, respectively. The conformational differences in the tetramethylene unit were discussed as main reasons for the difference in the fiber periods of the two crystalline modifications. Melt polycondensates were reported to have monoclinic unit cells.

For PBA, two crystallographic structures have been reported, too. The monoclinic $\alpha$-modification with $a = 0.673$ nm, $b = 0.794$ nm, $c = 1.420$ nm and $\beta = 45.5°$ is often accompanied by the orthorhombic $\beta$-modification with $a = 0.506$ nm, $b = 0.735$ nm and

$c = 1.467$ nm [79–84]. The content of β-phase depends strongly on the preparation conditions. Gan et al. [51,54] found higher contents of β-phase after isothermal crystallization at lower temperature (T = 25 °C), while crystallization at 35 °C gave only α-phase. This was found for samples that were dissolved, precipitated and isothermally crystallized. The monoclinic α-form was reported to be thermodynamically more stable. Comparable results were reported by Woo et al. [54,55].

Figure 5 shows the WAXS patterns of PBA and PBS samples obtained by enzymatic solution polycondensation in toluene (precipitated in methanol and dried), by enzymatic melt polycondensation (dissolved in chloroform, precipitated in methanol and dried) and by conventional melt polycondensation. The work-up procedure of the enzymatically synthesized samples resulted in roughly comparable solid state structures. All PBA samples (except PBA9) were synthesized at the same temperature (T = 70 °C). All samples show α- as well as β-modification in different extends, visible by the reflections (110) and (020) at different scattering positions, as indicated in Figure 5. In contrast to PBA, the PBS samples did not form the crystalline β-phase independent of the reaction conditions, which is in accordance to the literature reporting that the β-phase develops during strain stress [60]. The total crystallinity of all samples is higher than that of the PBA samples obtained under comparable conditions (see Table 3).

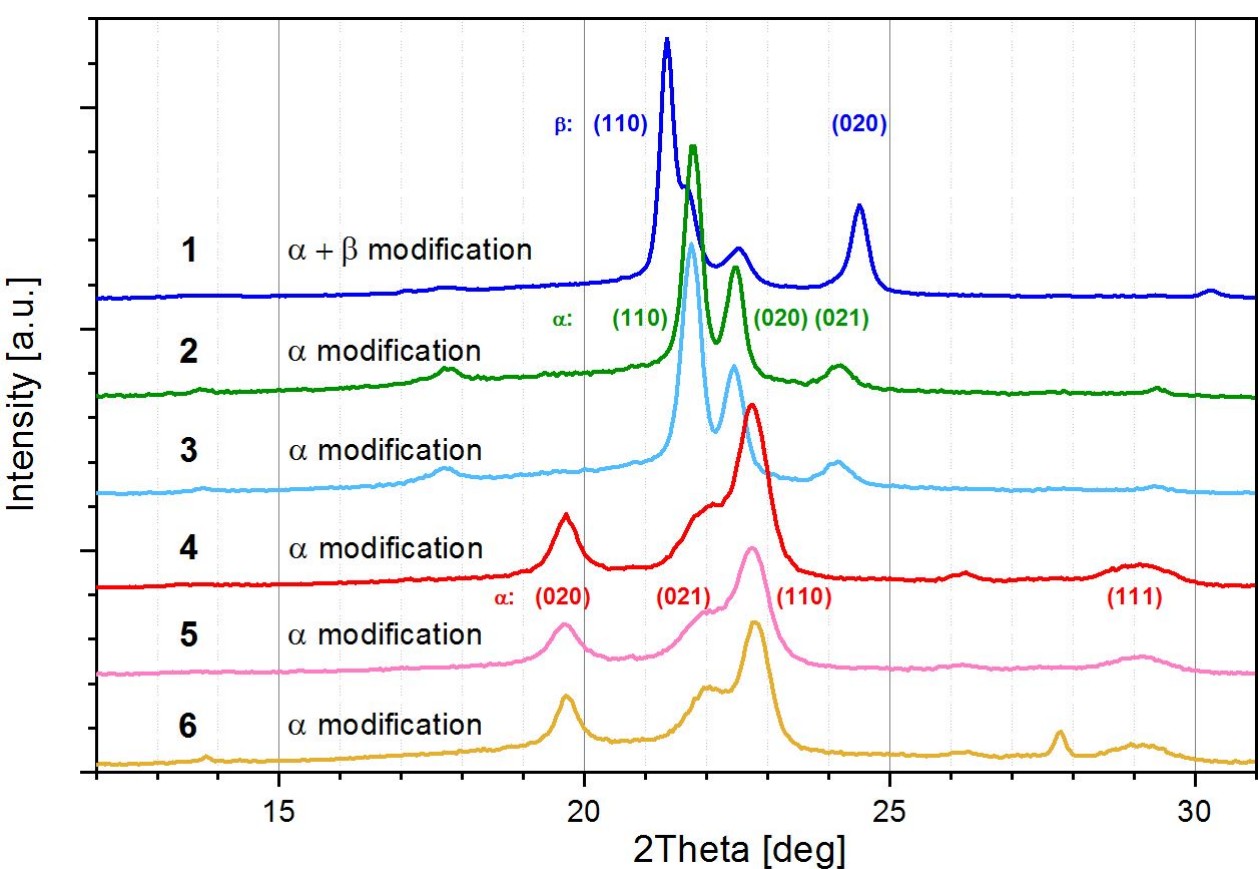

**Figure 5.** WAXS diffraction pattern of PBA and PBS samples obtained by enzymatic polycondensation with CALB in solution and in melt compared to samples from conventional melt polycondensation: (**1**) PBA4 in solution; (**2**) PBA7 in melt with CALB; (**3**) PBA9 in melt Ti(OBu)$_4$; (**4**) PBS5 in solution; (**5**) PBS8 in melt with CALB; (**6**) PBS1 in melt with Ti(OBu)$_4$.

The β-phase was always found in samples obtained from solution polycondensation. The lower the concentration of the reactants in solution the higher was the content of β-phase (PBA 5 vs. PBA 4). In solution polycondensates PBA4, PBA5, and PBA5-1, $c_β$ increased with increasing molar mass (compare Table 2). $c_β$ was significantly reduced in the samples obtained by melt polycondensation independent of the reaction temperature.

These samples all developed the thermodynamically more stable α-form. The results show that the polycondensation procedure and molar mass both have a significant influence on the solid state structure of as-synthesized PBA.

**Table 3.** X-ray crystallinities for selected PBA and PBS samples obtained by enzymatic polycondensation with CALB in solution and in melt.

| Polymer | Polycondensation Conditions | Total Crystallinity $c_{\alpha+\beta}$ | Crystallinity $c_\beta$ |
|---|---|---|---|
| PBA4 | Solution, 70 °C, CALB | 0.58 | 0.36 |
| PBA5 | Solution, 70 °C, CALB | 0.53 | 0.13 |
| PBA5-1 | Solution, 70 °C, CALB | 0.52 | 0.06 |
| PBA7 | Melt, 70 °C, CALB | 0.36 | 0.01 |
| PBA9 | Melt, 245 °C, Ti(OBu)$_4$ | 0.46 | 0.01 |
| PBS5 | Solution, 80 °C, CALB | 0.69 | -[1] |
| PBS7 | Melt, 80 °C, CALB | 0.74 | -[1] |
| PBS8 | Melt, 150 °C, CALB | 0.75 | -[1] |
| PBS1 | Melt, 245 °C, Ti(OBu)$_4$ (stirring autoclave) | 0.50 | -[1] |
| PBS1-2 | Melt, 245 °C, Ti(OBu)$_4$ (lab) | 0.57 | -[1] |

[1] β-phase not detectable.

The WAXS results were complemented by DSC measurements with a heating and cooling rate of ±10 K/min. The results are depicted in Figure 6 and reflect a picture comparable to the WAXS results. The presence of a high concentration of the β-form in PBA4 gives rise to two separated melting peaks in the 1st heating [79,85]. Such a state is usually formed after isothermal crystallization at lower temperature. In the cooling curve of the sample, only one crystallization peak was observed. In the 2nd heating run, the melting endotherms were positioned closer to the peaks observed for the melt polycondensates, suggesting the transformation from β- into the more stable α-phase occurred. The melt polycondensates with very low content of β-phase formed multiple melting peaks. The existence of multiple melting peaks has been described in the literature [79,85,86]. They were formed mainly after isothermal crystallization at intermediate temperatures and in mixtures of α- and β-phase. The samples studied here did not contain β-phase. Thus, the multiple endotherms have to be caused by differences in the crystallite structure (ringless and ring-banded spherulites [83]), which were again changed during the 2nd heating as visible by the shift of melt peak maxima.

The PBS samples (showing only the crystalline α-phase) also revealed influences of the polycondensation procedure. The highly crystalline solution polycondensates had one melt peak maximum with an adjacent shoulder at the flank to lower temperature, a pattern which was reported for PBS single crystals grown at higher temperature (39–62 °C), while the melt polycondensates showed a picture typical for single crystals obtained after isothermal crystallization at 30 °C [86]. This behavior reflects the different cooling procedures during the synthesis. PBA from solution polycondensation precipitated during synthesis (i.e., at 70 °C), while the melt polycondensate was cooled down slowly to room temperature in the flask. The behavior of the samples in the 2nd heating run is rather comparable. Before melting with again two melt peaks occurred, a recrystallization took place which is also known from literature [87].

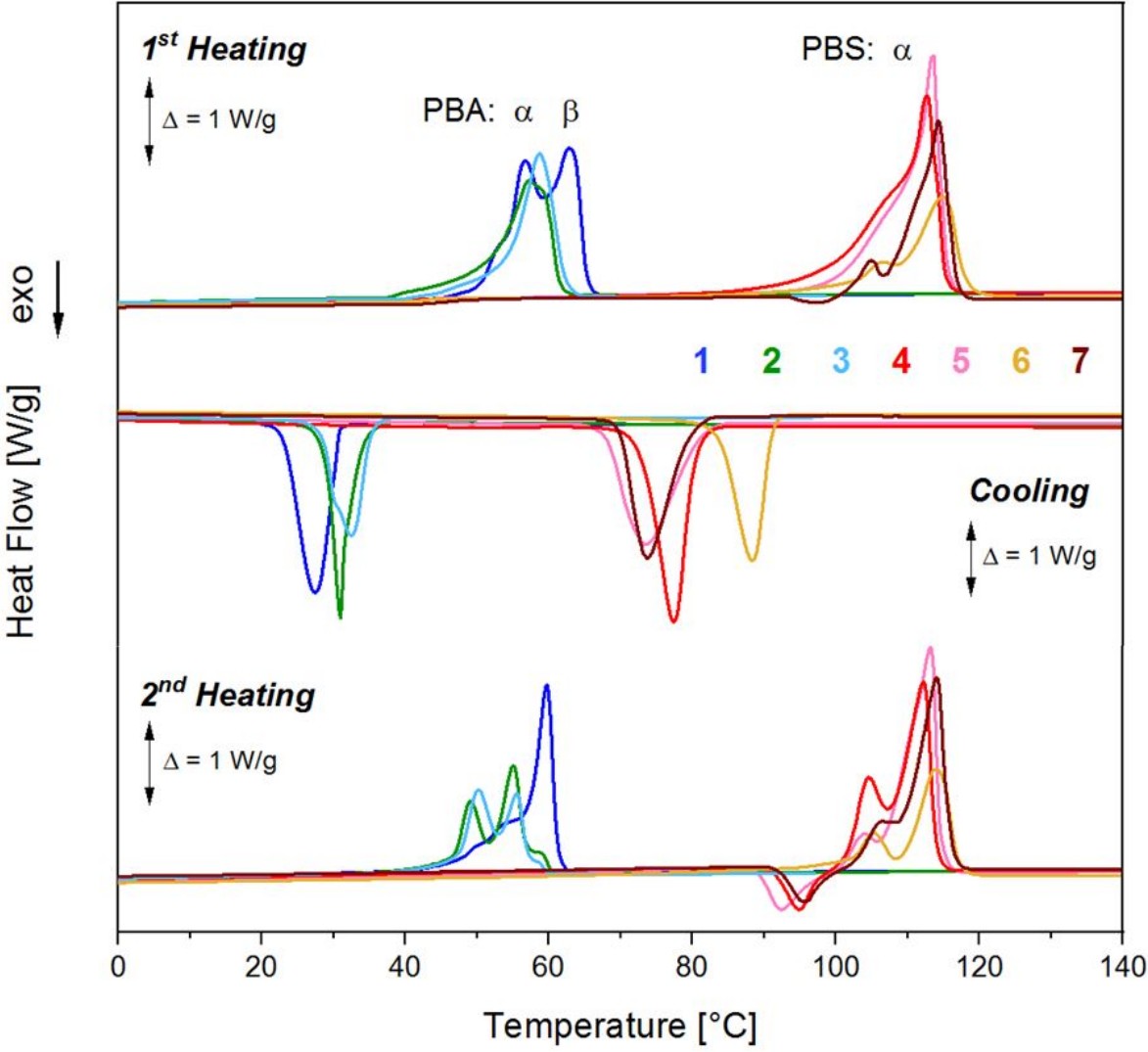

**Figure 6.** DSC curves (1st heating, cooling, 2nd heating) of PBA and PBS samples obtained by enzymatic polycondensation with CALB in solution and in melt compared to samples from conventional melt polycondensation: (**1**) PBA4 in solution; (**2**) PBA7 in melt with CALB; (**3**) PBA9 in melt Ti(OBu)$_4$; (**4**) PBS5 in solution; (**5**) PBS8 in melt with CALB; (**6**) PBS1 in melt with Ti(OBu)$_4$; and (**7**) PBS1-3 in melt with Ti(OBu)$_4$ with higher molar mass.

## 4. Conclusions

Polycondensation catalyzed by enzymes represents an eco-friendly alternative compared to the conventional organometal and metal oxide catalysis. Performing the reaction without solvent, in melt, further enhances the sustainability of the process, but can be limited by diffusion control. The enzyme CALB used is a renewable biocatalyst with high catalytic activity that can operate under mild reaction conditions. It could be demonstrated here that the immobilized form N435 was also active at temperatures up to 150 °C which is a pre-condition to perform melt polycondensation for PBS.

The influence of the reaction conditions on CALB-catalyzed polycondensations of both divinyladipate with 1,4-butanediol, and of dimethylsuccinate with 1,4-butanediol was studied. This allowed analyzing the effect of CALB activity and reaction conditions under comparable reaction conditions and with comparable analytical methods, on the relative molar mass and the solid state structure of the synthesized polyesters. The formed products were characterized in detail by a combination of SEC, $^1$H NMR, MALDI-TOF MS and WAXS analysis.

It was shown that the activity of polymer-immobilized CALB (N435) depended on the storage time of the material and directly influenced the relative molar mass achieved. Furthermore, we were able to demonstrate that the enzyme also worked in the diffusion-controlled melt polycondensation at unexpected high temperatures up to 150 °C (for PBS). CALB-catalyzed polycondensation in solution for 24 h at 70 °C proved to achieve PBA with molar masses higher than reported in the literature [18–20,61] up to $M_w$ of 60,000 g/mol, comparable to conventional PBA, while melt polycondensation resulted in a moderate decrease of molar mass to $M_w$ ~31,000. The molar masses of PBS achieved with CALB-catalyzed melt polycondensation were higher than those reported in the literature and were in the range of those reported for solution polycondensates [39,45,47]. They will be increased when the catalyst is not removed by dissolving/precipitation work-up procedures necessary for molar mass analysis. The reaction conditions showed an influence of the solid state structure. However, melting of the samples (as shown in the 2nd heating run of DSC) unifies the characteristics, made the properties comparable (only depending on molar mass) and reduced the crystallinity.

In summary, we showed that enzymatic polycondensation of PBA and PBS in melt worked well and reduced the reaction times necessary for polycondensation in solution ranging from 24 h up to several day drastically to three hours in vacuum. The molar masses can be further increased by up-scaling and further optimization of the melt polycondensation. We suggest for practical use not to remove the enzyme immobilized on a resin and use the material as it is without work-up. This will further enhance the sustainability of the whole process.

**Supplementary Materials:** The following are available online at https://www.mdpi.com/2227-9717/9/3/411/s1, Figure S1: Headspace GC/MS spectra of products found in the gas phase during polycondensation of divinyladipate with 1,4-butanediol (from top to bottom): (1) Gas phase product ($t$ = 1.58 s, acetaldehyde; $t$ = 3.68 s, toluene); (2) empty vial before measurement; (3) empty vial after measurement; Figure S2: $^1$H NMR spectrum of PBA4 (in CDCl$_3$); Figure S3: $^1$H NMR spectrum of PBA5-1 (in CDCl$_3$); Figure S4: $^1$H NMR spectrum of PBA6 (in CDCl$_3$); Figure S5: $^1$H NMR spectrum of PBA2 (in CDCl$_3$); Figure S6: $^1$H NMR spectrum of PBS1-2 (in CDCl$_3$); Figure S7: $^1$H NMR spectrum of PBS7 (in CDCl$_3$); Figure S8: $^1$H NMR spectrum of PBS5 (in CDCl$_3$).

**Author Contributions:** Conceptualization, D.P., R.C. and D.F.; methodology, D.P., R.C. and D.J.; validation, R.C., D.F. and A.K.; investigation, R.C., D.F., K.S., D.J., A.K. and P.F.; resources, P.F.; data curation, D.P., D.J. and A.W.; writing—original draft preparation, D.P. and D.J.; writing—review and editing, D.P., D.J., A.W. and B.V.; visualization, D.P., D.J. and P.F.; supervision, D.P. and B.V. All authors have read and agreed to the published version of the manuscript.

**Funding:** This research received no external funding.

**Institutional Review Board Statement:** Not applicable.

**Informed Consent Statement:** Not applicable.

**Data Availability Statement:** The Supporting Information is available under https://cloud.ipfdd.de/getlink/fiNicyxsYBZJKo9qrQDwVGhP/processes-1118631-supplementary.pdf?logInfo=Dateilink%20f%C3%BCr%20Datei%20processes-1118631-supplementary.pdf%20von%20Doris%20Pospiech%20gespeichert.

**Acknowledgments:** R. Choinska acknowledges financial support of Leibniz-Institut für Polymerforschung Dresden e.V. for two guest stays at IPF. Furthermore, the authors want to thank coworkers of IPF for their contributions: H. Komber for NMR measurements and discussion, P. Treppe for SEC measurements, and K. Arnhold for DSC and TGA measurements.

**Conflicts of Interest:** The authors declare no conflict of interest.

## Abbreviations

| | |
|---|---|
| CALB | *Candida antarctica* Lipase B |
| PBA | poly(butylene adipate) |
| PBS | poly(butylene succinate) |
| N435 | Novozyme 435, immobilized lipase B from *Candida Antarctica,* produced by Novozymes (DK) |
| Ti(OBu)$_4$ | titanium(IV)n-butoxide |
| Sb$_2$O$_3$ | antimony trioxide |
| BD | 1,4-butanediol |
| DMS | dimethyl succinate |
| DMA | dimethyl adipate |
| SEC | size exclusion chromatography |
| NMR | nuclear magnetic resonance |
| MALDI-TOF MS | matrix-assisted laser desorption ionization–time-of-flight mass spectrometry |
| WAXS | wide-angle X-ray scattering |
| DSC | differential scanning calorimetry |
| $M_w$ | weight average molar mass |
| $M_n$ | number average molar mass |

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
