# Peer review of "Enzymatic Synthesis of Poly(alkylene succinate)s: Influence of Reaction Conditions"

_processes, doi:10.3390/pr9030411_

Round 1

Reviewer 1 Report

This manuscript looks good, and the results are interesting. However, the authors should apply the following comments. So, the manuscript can be accepted after minor revision.

Author Response

Processes

Manuscript processes-1118631

Pospiech et al.

Title: Enzymatic synthesis of poly(alkylene succinate)s: Influence of reaction conditions

First of all, we would like to thank the reviewers and editors for taking their time to evaluate the manuscript. We appreciate this a lot, took all comments carefully into account and revised the manuscript accordingly.

All replies are given in red right below the comment.

Reviewer 1:

This manuscript looks good, and the results are interesting. However, the authors should apply the following comments. So, the manuscript can be accepted after minor revision.

Introduction

  1. After the following sentences should be the corresponding references (page 3, line 100-103):.

Biodegradability, relatively good mechanical properties and processability make these aliphatic polyesters useful for variety of industrial, medical, and agricultural applications. PBS has found widespread applications in different areas ranging from food packaging, fiber/textile, electrics and electronics, agricultural mulch films etc.

Thank you. Some relevant references were inserted.

  1. The following parts should be written relatively shorter.

- page 3 Line 66-91;

Thank you. We tried to shorten the complete Introduction (in particular the parts mentioned by reviewer 1) and re-organized it. We hope that it is now better suited.

- page 3-4 Line 105-135.

Thus, introduction section should be re-written according the abovementioned comments.

Done, see above.

Results and Discussion

  1. After the following sentences should be the corresponding references (page 8, line 335-337):

In the discussion of the results and especially in the comparison with literature results it has to be taken into account that the molar masses are relative values and may also depend on the analysis conditions.

We cannot give a reference for that. It is concluded from our own work of more than 35 years of polycondensation of a variety of polyesters. All of these were analyzed by SEC (GPC) resulting in relative molar masses, and it happened more than one time that results were not completely comparable. This is a general knowledge in all labs working with SEC of polyesters. We also did extensive comparisons with other labs using the same samples and obtained different results.

The determination of absolute molar masses requires either MALLS detection (which cannot be applied under the conditions used here with pentafluorphenol/chloroform as eluent) or the knowledge of the constants of the respective Kuhn-Mark-Houwink-Sakurada equation (which are different for each polymer).

For all these reasons we would like to ask not to give references here.

Reviewer 2 Report

I appreciate the opportunity to contribute reviewing the manuscript # processes-1118631 entitled, “Enzymatic synthesis of poly(alkylene succinate)s: Influence of reaction conditions” by Pospiech et al.

This manuscript addresses polycondensation of aliphatic polyesters by using lipases as catalyst.  The authors have a great perspective because the enzyme-catalyzed polycondensation in melt state has been poorly studied so far. The overall research design and characterization of the resulting polyesters are fine. Experimental results are okay in most places. There are some minor issues in this manuscript should be addressed, and therefore, minor revision is recommended.

Comments:

  1. In line 360,the authors states that “Slight excess of 1,4-butanediol enhanced the molar mass” However, in a general polycondensation reaction, the equimolarity of the two monomers is important for the molecular weight of the produced polymer. Why did this result, and is there any possibility?

  1. The crystallinity of the produced polymers is compared using XRD and DSC measurements. In DSC measurement, the authors are discussing using 1st cooling and 2nd heating, but it is probably 1st heating that directly reflects the characteristics of the product crystallinity. Once the polymer is melted, there is no hysteresis of the formation process. I think it is better to discuss DSC measurement based on 1st heating.

I have some other comments.

  1. Please check out the many "Error! Reference source not found" in line 61, 188, 189, 200, 313, 322, 323, 334, 378, 380, 406, 424, 429, 482, 489, 494, 499, 511.

  1. Line 24; “poly(butylene adipate) and poly(butylene succinate)” --> “poly(butylene adipate) (PBA) and poly(butylene succinate) (PBS)”

Author Response

Processes

Manuscript processes-1118631

Pospiech et al.

Title: Enzymatic synthesis of poly(alkylene succinate)s: Influence of reaction conditions

First of all, we would like to thank the reviewers and editors for taking their time to evaluate the manuscript. We appreciate this a lot, took all comments carefully into account and revised the manuscript accordingly.

All replies are given in red right below the comment.

Reviewer 2:

This paper investigates application of lipases for melt polycondensation of aliphatic polyesters by transesterification of activated dicarboxylic acids with diols. This study demonstrates that Candida antarctica Lipase B successfully catalyzes polycondensation of  divinyladipate and dimethylsuccinate,  with 1,4-butanediol.

reviews [15,16] are not recent - 5 years old...

We inserted more recent references in the sentence, but also generally during the whole manuscript several more recent citations.

The sentence was reformulated: “A comprehensive overview on enzyme types useful for catalysis of esterifications, among them oxidoreductases like laccase and horseradish peroxidase, transferases, hydrolases (in particular lipases, ligases, papain, trypsin, a-chymotrypsin and many others is provided in several reviews [15,16,24,27,28].”

  1. 55 horseradish peroxidase

Corrected, thank you!

in whole manuscript - full of "errors - reference not found" ... you must doublecheck your generated pdf file prior to its submission!

We are very sorry for that. We checked the pdf of the manuscript before submission and did not found that. For submission of the final manuscript, all links will be removed, and therefore, this should not happen again. Er checked the pdf version of the final revised manuscript and it appears allright.

  1. 60 you cannot start sentence with vice versa – rephrase

We changed the sentence into: “In the converse, it can also catalyze esterification.”

  1. 66-90 - this is full of well known, textbook statements - please, condense it and remove banal, generally known points (as e.g. about the catalytic triad). 

              Thank you. We shortened the Introduction and re-organized it, see above.

  1. 113-114  rephrase, do not start sentence with ... please note ... (this is not a letter)

The sentence was changed as following:

“The relative molar masses determined under different lab conditions should not be compared directly, the values only give a rough trend.”, please also note reply to reviewer 1

  1. 26 ..achieves...

Corrected, thank you.

  1. 27    It is not a standard to cite references in abstract (by the numbers) - e.g. ref. 1-4. Abstract must be fully autonomous as it is often published in secondary literature without direct access to the reference list. Please, kindly amend this situation. Moreover, refs. 1-4 must be cited at least in introduction (or elsewhere).

Thank you. You are of course completely right, the abstract was corrected as suggested.

reference 5 is absolutely unsuitable, it does not deal with general properties and advantages of enzymes and namely lipases. Moreover series of Adv. Polym. Sci. are books and they are not easily accessible. Please, find some suitable and modern (not 15 y old) reference in some journal. You may wish to replace reference from this book with modern (sic!) and easily reachable references.

We removed ref. 5. As mentioned before, we added a number of very recent papers throughout the whole manuscript to improve its quality. Papers from Adv. Polym. Sci. have a DOI and can easily be accessed (it was no problem to get the pdfs).

However, just as remark: we believe that the old references already represent a high standard of knowledge in enzyme-catalyzed polycondensation that should not be neglected.

MatMet

pls. add to all producers town and country

Done, thank you.

  1. 218 p-nitrophenyl palmitate (para - italics - and elsewhere), italicize also names of organisms (e.g. ref. 22,31,36 and elsewhere), ref. 36 spelling

Done, thank you!

all references - pls make homogenous writing capitals in  paper titles  - preferably all small letters.

We gave the references as they appear in the papers and journals. Therefore, we did not change it at present (thinking that original references should not be changed ??). The editors may decide during the further processing if this should be done or not. Of course, we removed writing mistakes (soryr for that, originated by the citation tool used).

There is lot of abbreviations in the text, some of them not standard ones, pls. add a list of abbreviations.

We introduced each abbreviation in the text for using it and added additionally a list of abbreviations at the end.

Pls. doublecheck the whole manuscript for typos and use some spellchecker to remove some basic inconsistencies.

Done.

In general this paper has a merit and after revision it will be publishable.

One again, we want to thank the reviewers and hope that the corrections done improved the quality of the manuscript. We are grateful for publication of our results. Please do not hesitate to contact us for further questions.

Sincerely,

Dr. Doris Pospiech

(on behalf of all coauthors)

Reviewer 3 Report

This paper investigates application of lipases for melt polycondensation of aliphatic polyesters by transesterification of activated dicarboxylic acids with diols. This study demonstrates that Candida antarctica Lipase B successfully catalyzes polycondensation of  divinyladipate and dimethylsuccinate,  with 1,4-butanediol.

reviews [15,16] are not recent - 5 years old...

l. 55 horseradish peroxidase

in whole manuscript - full of "errors - reference not found" ... you must doublecheck your generated pdf file prior to its submission!

l. 60 you cannot start sentence with vice versa - rephrase

ll. 66-90 - this is full of well known, textbook statements - please, condense it and remove banal, generally known points (as e.g. about the catalytic triad). 

l. 113-114  rephrase, do not start sentence with ... please note ... (this is not a letter)

l. 26 ..achieves...

l. 27    It is not a standard to cite references in abstract (by the numbers) - e.g. ref. 1-4. Abstract must be fully autonomous as it is often published in secondary literature without direct access to the reference list. Please, kindly amend this situation. Moreover, refs. 1-4 must be cited at least in introduction (or elsewhere).

reference 5 is absolutely unsuitable, it does not deal with general properties and advantages of enzymes and namely lipases. Moreover series of Adv. Polym. Sci. are books and they are not easily accessible. Please, find some suitable and modern (not 15 y old) reference in some journal. You may wish to replace reference from this book with modern (sic!) and easily reachable references.

MatMet

pls. add to all producers town and country

l. 218 p-nitrophenyl palmitate (para - italics - and elsewhere), italicize also names of organisms (e.g. ref. 22,31,36 and elsewhere), ref. 36 spelling

all references - pls make homogenous writing capitals in  paper titles  - preferably all small letters.

There is lot of abbreviations in the text, some of them not standard ones, pls. add a list of abbreviations.

Pls. doublecheck the whole manuscript for typos and use some spellchecker to remove some basic inconsistencies.

In general this paper has a merit and after revision it will be publishable.

Author Response

Processes

Manuscript processes-1118631

Pospiech et al.

Title: Enzymatic synthesis of poly(alkylene succinate)s: Influence of reaction conditions

First of all, we would like to thank the reviewers and editors for taking their time to evaluate the manuscript. We appreciate this a lot, took all comments carefully into account and revised the manuscript accordingly.

All replies are given in red right below the comment.

Reviewer 3:

This paper investigates application of lipases for melt polycondensation of aliphatic polyesters by transesterification of activated dicarboxylic acids with diols. This study demonstrates that Candida antarctica Lipase B successfully catalyzes polycondensation of  divinyladipate and dimethylsuccinate,  with 1,4-butanediol.

reviews [15,16] are not recent - 5 years old...

We inserted more recent references in the sentence, but also generally during the whole manuscript several more recent citations.

The sentence was reformulated: “A comprehensive overview on enzyme types useful for catalysis of esterifications, among them oxidoreductases like laccase and horseradish peroxidase, transferases, hydrolases (in particular lipases, ligases, papain, trypsin, a-chymotrypsin and many others is provided in several reviews [15,16,24,27,28].”

  1. 55 horseradish peroxidase

Corrected, thank you!

in whole manuscript - full of "errors - reference not found" ... you must doublecheck your generated pdf file prior to its submission!

We are very sorry for that. We checked the pdf of the manuscript before submission and did not found that. For submission of the final manuscript, all links will be removed, and therefore, this should not happen again. Er checked the pdf version of the final revised manuscript and it appears allright.

  1. 60 you cannot start sentence with vice versa – rephrase

We changed the sentence into: “In the converse, it can also catalyze esterification.”

  1. 66-90 - this is full of well known, textbook statements - please, condense it and remove banal, generally known points (as e.g. about the catalytic triad). 

              Thank you. We shortened the Introduction and re-organized it, see above.

  1. 113-114  rephrase, do not start sentence with ... please note ... (this is not a letter)

The sentence was changed as following:

“The relative molar masses determined under different lab conditions should not be compared directly, the values only give a rough trend.”, please also note reply to reviewer 1

  1. 26 ..achieves...

Corrected, thank you.

  1. 27    It is not a standard to cite references in abstract (by the numbers) - e.g. ref. 1-4. Abstract must be fully autonomous as it is often published in secondary literature without direct access to the reference list. Please, kindly amend this situation. Moreover, refs. 1-4 must be cited at least in introduction (or elsewhere).

Thank you. You are of course completely right, the abstract was corrected as suggested.

reference 5 is absolutely unsuitable, it does not deal with general properties and advantages of enzymes and namely lipases. Moreover series of Adv. Polym. Sci. are books and they are not easily accessible. Please, find some suitable and modern (not 15 y old) reference in some journal. You may wish to replace reference from this book with modern (sic!) and easily reachable references.

We removed ref. 5. As mentioned before, we added a number of very recent papers throughout the whole manuscript to improve its quality. Papers from Adv. Polym. Sci. have a DOI and can easily be accessed (it was no problem to get the pdfs).

However, just as remark: we believe that the old references already represent a high standard of knowledge in enzyme-catalyzed polycondensation that should not be neglected.

MatMet

pls. add to all producers town and country

Done, thank you.

  1. 218 p-nitrophenyl palmitate (para - italics - and elsewhere), italicize also names of organisms (e.g. ref. 22,31,36 and elsewhere), ref. 36 spelling

Done, thank you!

all references - pls make homogenous writing capitals in  paper titles  - preferably all small letters.

We gave the references as they appear in the papers and journals. Therefore, we did not change it at present (thinking that original references should not be changed ??). The editors may decide during the further processing if this should be done or not. Of course, we removed writing mistakes (soryr for that, originated by the citation tool used).

There is lot of abbreviations in the text, some of them not standard ones, pls. add a list of abbreviations.

We introduced each abbreviation in the text for using it and added additionally a list of abbreviations at the end.

Pls. doublecheck the whole manuscript for typos and use some spellchecker to remove some basic inconsistencies.

Done.

In general this paper has a merit and after revision it will be publishable.

One again, we want to thank the reviewers and hope that the corrections done improved the quality of the manuscript. We are grateful for publication of our results. Please do not hesitate to contact us for further questions.

Sincerely,

Dr. Doris Pospiech

(on behalf of all coauthors)

Round 2

Reviewer 3 Report

Authors addressed all points and the paper is now recommended for acceptance.